# Metabolism-Related Gene Pairs to Predict the Clinical Outcome and Molecular Characteristics of Early Hepatocellular Carcinoma

**DOI:** 10.3390/cancers14163957

**Published:** 2022-08-16

**Authors:** Junling Wu, Zeman Lin, Daihan Ji, Zhenli Li, Huarong Zhang, Shuting Lu, Shenglin Wang, Xiaolong Liu, Lu Ao

**Affiliations:** 1Fujian Key Laboratory of Medical Bioinformatics, Department of Bioinformatics, School of Medical Technology and Engineering, Fujian Medical University, Fuzhou 350025, China; 2The United Innovation of Mengchao Hepatobiliary Technology Key Laboratory of Fujian Province, Mengchao Hepatobiliary Hospital of Fujian Medical University, Fuzhou 350025, China; 3Key Laboratory of Ministry of Education for Gastrointestinal Cancer, School of Basic Medical Sciences, Fujian Medical University, Fuzhou 350122, China

**Keywords:** metabolism-related genes, hepatocellular carcinoma, risk stratification model, prognosis, molecular characteristics

## Abstract

**Simple Summary:**

After surgery, about 60–70% of early hepatocellular carcinoma patients suffer from relapse within 5 years, hindering long-term survival. Clinical and pathologic features cannot provide an accurate evaluation. We aimed to construct a stratification model from the metabolic aspect to predict the clinical outcome and reveal the molecular characteristics of different prognostic subgroups. An individualized metabolic signature of 10 gene pairs was developed from 250 early HCCs and validated in 311 samples from different datasets. The signature stratified early HCC cases one-by-one into two risk groups with different survival rates. The molecular characteristics of the two risk groups were analyzed by multi-omics data. The relationships with proliferation, immunity, and drug benefits were summarized. The signature was further validated in 47 institutional transcriptomic HCC samples and 101 public proteomic samples.

**Abstract:**

Recurrence is the main factor affecting the prognosis of early hepatocellular carcinoma (HCC), which is not accurately evaluated by clinical indicators. The metabolic heterogeneity of HCC hints at the possibility of constructing a stratification model to predict the clinical outcome. On the basis of the relative expression orderings of 2939 metabolism-related genes, an individualized signature with 10 metabolism-related gene pairs (10-GPS) was developed from 250 early HCC samples in the discovery datasets, which stratified HCC patients into the high- and low-risk subgroups with significantly different survival rates. The 10-GPS was validated in 311 public transcriptomic samples from two independent validation datasets. A nomogram that included the 10-GPS, age, gender, and stage was constructed for eventual clinical evaluation. The low-risk group was characterized by lower proliferation, higher metabolism, increased activated immune microenvironment, and lower TIDE scores, suggesting a better response to immunotherapy. The high-risk group displayed hypomethylation, higher copy number alterations, mutations, and more overexpression of immune-checkpoint genes, which might jointly lead to poor outcomes. The prognostic accuracy of the 10-GPS was further validated in 47 institutional transcriptomic samples and 101 public proteomic samples. In conclusion, the 10-GPS is a robust predictor of the clinical outcome for early HCC patients and could help evaluate prognosis and characterize molecular heterogeneity.

## 1. Introduction

Liver cancer is the third-leading cause of cancer death [1]; 90% of cases are primary hepatocellular carcinoma (HCC) [2]. Surgical resection is the first choice of treatment for early HCC patients whose liver function is well-preserved. Nevertheless, approximately 60–70% of patients suffer from relapse within 5 years. The high recurrence rate after resection has become an important factor hindering the long-term survival of patients [3]. Clinical and pathologic features, such as the tumor–node–metastasis (TNM) stage, alpha-fetoprotein (AFP) level, and abnormal thrombin, cannot provide an accurate evaluation of clinical outcomes in HCC patients [4,5]. Moreover, liver cancer is considered a metabolic disease with high heterogeneity [6,7] that is selectively advantageous for tumor growth, proliferation, and survival [8]. The distinct metabolism of HCC patients hints at the possibility of constructing a stratification model from the metabolic aspect to predict the clinical outcome.

Previously reported prognostic signatures for HCC are based on risk scores obtained from the expression levels of marker genes [6,9,10], which are susceptible to batch effects. A batch of samples needs to be collected in advance and to be normalized together for the application of such signatures, which is not in accord with clinical practice. Moreover, normalizing samples together would have an impact on the risk classification of a patient [11]. By contrast, it has been demonstrated that the relative expression orderings (REOs) of genes within samples are resistant to batch effects [12] and are robust across various platforms with different designs [13]. More notably, within-sample REOs can provide personalized judgment for a single sample without data normalization. A sample is individually classified according to the REOs of gene pairs in the signature [14,15,16], which is more in line with actual clinical needs.

Using 2939 metabolic genes encoding human metabolic enzymes and small-molecule transporters, an individually prognostic signature was developed from 250 stage I–II HCC patients and validated in 311 transcriptomic samples from four public datasets. Then, multivariate Cox regression analysis and nomogram establishment with clinical factors were performed. The functional and clinical characteristics, epigenomic and genomic alterations, immune infiltration, and therapeutic benefits of the two prognostic groups were analyzed. The 10-GPS was further validated in 47 institutional transcriptomic samples and 101 public proteomic samples. By relating the expressions of metabolic genes with the prognosis of early HCC, the metabolism-related signature developed in this study can individually stratify patients, which will improve their prognosis, provide biological insights, predict therapeutic benefits, and assist in clinical management.

## 2. Materials and Methods

### 2.1. Study Selection Criteria

Publications that explored the prognosis of HCC and were with sufficient and available transcriptomic data and prognostic data, such as the progression-free interval (PFI, time from primary treatment to disease recurrence), disease-free survival (DFS, time from surgery to tumor recurrence, distant metastasis, or death), and overall survival (OS, time from surgery to death) were included in this study. The survival times were defined as described in the original publications [17,18,19,20]. Only TNM stage I–II HCC samples with resection were collected and analyzed.

### 2.2. Data Collection and Preprocessing

In total, 197 samples from The Cancer Genome Atlas [21] (TCGA, version 26.0, https://portal.gdc.cancer.gov/, accessed on 27 October 2020), 53 samples in GSE116174 and 170 samples in GSE14520 from Gene Expression Omnibus (GEO, http://www.ncbi.nlm.nih.gov/geo/, accessed on 27 October 2020), and 141 samples from International Cancer Genome Consortium [19] (ICGC, version 28, https://dcc.icgc.org/projects/LIRI-JP/, accessed on 27 October 2020) were downloaded. All datasets were renamed as HCC with the sample size, e.g., HCC197. The demographic and clinicopathologic characteristics of samples are described in Table 1. Datasets HCC197 and HCC53 were used to identify the signature; HCC170 and HCC141 were used for independent validation. The raw count represented the gene expression value of the RNA-Seq samples. The robust multiarray average algorithm was used to process the raw mRNA expression data (.CEL) from a microarray [22]. The probe ID was matched with the Entrez gene ID. One probe that matched multiple genes was deleted. The mean expression value of a gene was taken if the gene was mapped by multiple probes.

Proteomic data and clinical information of 101 stage I–II HCC patients, denoted as HCC101 (Table 1), were downloaded from the clinical proteomic tumor analysis consortium (CPTAC) [23] database. These data were used to validate the signature at the proteomic level, which might facilitate its routine implementation in clinical settings.

### 2.3. Samples Collection and Data Measurement

Primary tumor tissues from 72 institutional HCC patients were collected during surgery operations at Mengchao Hepatobiliary Hospital of Fujian Medical University. Primary HCC was diagnosed by at least two experienced pathologists. No patient received treatment before surgical resection. All participants signed informed consent before enrolment. All study protocols were approved by the Institution Review Board of Mengchao Hepatobiliary Hospital of Fujian Medical University and performed under the Helsinki Declaration.

RNA extracted from 72 HCC samples was subjected to whole-transcriptome sequencing on an Illumina HiSeq ×10 platform (paired-end, 150 bp) by Annoroad Gene Tech. (Beijing, China) Co., Ltd. These institutional transcriptomic data of 47 patients at stage I–II, denoted as HCC47, were used to further validate the signature, and the raw counts of the genes in the RNA-Seq profiles were analyzed. The relapse-free survival (RFS, time from surgery to disease recurrence or death) and clinical information are shown in Table 1. Institutional transcriptomic data are available in the Genome Sequence Archive (GSA) repository (GSA accession number: HRA000464). More detailed information can be found in our previous work [24].

### 2.4. Multi-Omics Data of TCGA Portal

The multi-omics data of samples in the HCC197 dataset from TCGA were used to investigate the distinctive epigenomic and genomic characteristics between the high-risk and the low-risk subgroups. DNA methylation profiles were directly obtained from the UCSC Xena portal [20] (https://xenabrowser.net/datapages/, accessed on 27 November 2020). Probes with any “NA”-masked data points designed for sequences on sex chromosomes were deleted. CpG sites within the promoter regions with zero expression in more than 80% of the samples were deleted. In total, 9936 genes mapped by 17,028 CpG sites were analyzed. Somatic mutation data and copy number variation (CNV) were obtained from the GDC Data Portal (version 27.0, https://portal.gdc.cancer.gov/, accessed on 27 November 2020). For CNV data, the significant regions of gain or loss were identified by GISTIC 2.0 [25].

### 2.5. Identification of the Prognostic Signature with Metabolism-Related Gene Pairs

A list of 2939 metabolism-related genes was integrated from a previous study [26] and the latest metabolic pathways in the Kyoto Encyclopedia of Genes and Genomes [27] (KEGG) (version 97.0, http://www.genome.jp/kegg/kegg1.html, accessed on 15 January 2021) (Appendix A). The univariate Cox proportional-hazards regression model was used to find the candidate genes or gene pairs that were significantly correlated with the clinical outcome of early HCC. A gene whose expression level was significantly correlated with PFI in the HCC197 dataset was defined as a prognosis-associated gene. Two prognosis-associated genes, *G*_1_ and *G*_2_, with expression levels of *E*_1_ and *E*_2_, respectively, were constructed into a gene pair. Then, the REO pattern of the gene pair (*E*_1_ < *E*_2_ or *E*_1_ > *E*_2_) divided the samples into two groups. Gene pairs that were significantly associated with PFI were defined as prognosis-associated gene pairs. Then, all prognosis-associated gene pairs were ranked in descending order of the concordance index (C-index). Every prognosis-associated gene pair was selected as the seed, and a forward selection procedure starting with the first gene pair was performed to obtain the optimal subset that reached the highest C-index in dataset HCC197 and obtained a significant *p*-value of OS in dataset HCC53. The optimal subset of gene pairs was selected as the prognostic signature. The half-voting rule was used to decide the classification: a patient was classified into the high-risk group when at least half of the gene pairs in the prognostic signature voted for high risk; otherwise, the patient was classified into the low-risk group.

### 2.6. Survival Analysis and Differential Analysis

The multivariate Cox proportional hazards regression model was used to compute the independent prognostic value of the prognostic signature after adjusting for clinical factors. Survival curves were calculated by the log-rank test and visualized with the Kaplan–Meier plot. A nomogram with the signature, age, gender, and stage was constructed, and the 1-, 2-, and 3-year survival probability in overall samples was predicted. EdgeR and Student’s *t*-test were used to identify differentially expressed genes (DEGs) in the RNA-Seq count profiles and microarray profiles, respectively. The Wilcoxon rank-sum test was used to identify significant differential methylation (DM) sites between two risk groups. Fisher’s exact test was used to identify the differential frequencies of copy number alteration and mutation. Functional enrichment analysis of DEGs between the two risk groups was performed using the Database for Annotation, Visualization, and Integrated Discovery online tool (DAVID, version 6.8, https://david.ncifcrf.gov/, accessed on 15 January 2021). The Benjamini and Hochberg (BH) procedure was used to control the false discovery rate (FDR).

### 2.7. Statistical Analysis

Fisher’s exact test was performed to compare the distribution of samples classified by clinical factors or the 10-GPS. Eight representative proliferation genes obtained from the study by Désert et al. [28] were used to estimate the proliferation differences in two prognostic groups (Appendix A). The median score of the principal component analysis (PCA) of the eight genes was defined as the cut-off. A sample was assigned to the non-proliferative periportal phenotype (PP) if the risk score was greater than the cut-off value; otherwise, it was assigned to the proliferative HCC subclass. The average expression levels of 43 proliferation-associated genes were also used to calculate the proliferation scores [29] (Appendix A). Moreover, 108 metabolic pathways [30] with DEGs were used to explore the metabolic differences between the two prognostic groups. The metabolic activity was scored by gene set variation analysis (GSVA) using DEGs in each pathway, and the difference was compared by limma (Appendix A). The 29 immune-associated cell types [31] calculated by single-sample GSEA (ssGSEA) and 48 immune checkpoint genes [32] were used to estimate the different immune characteristics of the two prognostic groups. Tumor immune dysfunction and exclusion (TIDE) was calculated online (http://tide.dfci.harvard.edu/, accessed on 15 August 2021). A high TIDE score suggested that the patients were less likely to benefit from immune checkpoint inhibition (ICI) therapy [33]. Wilcoxon rank-sum tests were performed to compare these differences. The pRRophetic algorithm was used to evaluate the 50% inhibitory concentration (IC_50_) values of 138 antitumor drugs (including chemotherapy and targeted therapy) and identify potentially effective agents for patients in the two groups [34]. A *p*-value < 0.05 was considered significant for all analyses in this study unless otherwise stated. All statistical analyses were performed using R 3.6.3.

## 3. Results

### 3.1. Development and Validation of the Metabolism-Related Gene Pairs for Risk Stratification in Public Transcriptomic Datasets

The general flowchart of this study is described in Figure 1A. We first compared the sample distribution classified by available clinical factors in four transcriptomic datasets. The results showed that most clinical features among datasets were significantly different, suggesting the diversity of early HCC after surgery (Table 2). We then developed the signature to predict the clinical outcome of early HCC on the basis of the metabolism-related genes. In the HCC197 dataset, 164 metabolism-related genes whose expression levels were significantly correlated with the PFI of HCC patients were identified as prognosis-associated genes (FDR < 20%). Two of the 164 prognosis-associated genes were constructed into a gene pair. In total, 766 prognosis-associated gene pairs whose REOS were significantly correlated with PFI (*p* < 0.05) were identified. Then, the 10 gene pairs with the highest C-index values (C-index = 0.6933) in the HCC197 dataset and a significant *p*-value of OS in the HCC53 dataset, denoted as 10-GPS, were selected as the final risk stratification signature (Figure 1B). According to the half-voting rule, a patient was classified into the high-risk group if at least five gene pairs voted for high risk; otherwise, the patient was classified into the low-risk group. The 197 samples were separately classified into the high- and low-risk groups with 64 and 133 samples. The survival analysis showed that patients in the low-risk group had significantly longer PFI and OS than those in the high-risk group (PFI: HR = 4.35, 95% CI: 2.76–6.85, *p* = 6.17 × 10^−12^, C-index = 0.6933; OS: HR = 3.94, 95% CI: 2.22–6.99, *p* = 4.48 × 10^−7^, C-index = 0.6627) (Figure 1C). In the HCC53 dataset, 34 and 19 samples were separately stratified into the high- and low-risk groups. Similarly, the OS of the latter was significantly longer than that of the former (HR = 2.98, 95% CI: 1.00–8.86, *p* = 0.039, C-index = 0.6173, Figure 1D).

The 10-GPS was validated on two independent datasets that included 311 stage I–II samples. Consistent with the results observed in the discovery datasets, the survival analysis showed that the low-risk group had a significantly longer OS in the HCC141 dataset (HR = 2.93, 95% CI: 1.04–8.22, *p* = 0.033, C-index = 0.6992, Figure 1E) and significantly better DFS and OS in the HCC170 dataset (DFS: HR = 1.65, 95% CI: 1.08–2.52, *p* = 0.019, C-index = 0.5638; OS: HR = 2.41, 95% CI: 1.40–4.167, *p* = 0.001, C-index = 0.5990) (Figure 1F). Univariate Cox regression analysis showed that the 10-GPS and the stage were significantly associated with the prognosis of early HCC, and multivariate Cox regression analysis confirmed that the 10-GPS remained an independent predictor after adjusting for clinical variables, including stage, histologic grade, vascular invasion, AFP, and Child–Pugh grade of cirrhosis (Figure 1G).

In addition, the univariate and multivariate Cox regression analysis showed that stage was an important factor in a patient’s prognosis. Then, all samples in the four datasets were integrated, and the population was split into stages. The results showed that both stage I and stage II patients were divided by the 10-GPS into two prognostic groups with significantly different OS. Specifically, significantly more patients were predicted to be at high risk in stage II than in stage I (Chi-squared test, adjusted *p* = 5 × 10^−6^, Figure 2A), indicating that stage II HCC patients were more likely to be classified into the high-risk group. Furthermore, vascular invasion is also a significant factor for HCC prognosis. Patients with and without vascular invasion in HCC197 were divided by the 10-GPS into two prognostic groups with significantly different OS (Appendix A). No difference in vascular invasion was found between the distinct prognostic groups (Chi-squared test, adjusted *p* = 0.43). These results showed that the 10-GPS was associated with the OS of early HCC independent of stage and vascular invasion.

Moreover, we assessed the 3 common clinical variables in the overall series of HCC patients, which included gender, age, and stage, to develop a composite prognostic predictor of the 10-GPS. The nomogram showed the contribution of each variable to predict 1-year, 2-year, and 3-year survival probability (Figure 2B). Patients divided by the median of the composite score had significantly different survival (Figure 2C).

Conclusively, these results highlighted the robustness of the 10-GPS to stratify stage I–II HCC samples from different types of data into high- and low-risk groups with significantly different prognoses.

### 3.2. Distinct Proliferation and Metabolism Characteristics between the Two Prognostic Groups

With 10% FDR control, 3678 and 1777 DEGs were identified between the high- and low-risk groups in the HCC197 and HCC 141 datasets (edgeR, log2|Fold Change (FC)| > 1), while 20 and 1810 DEGs were identified between the two prognostic groups from the HCC53 and HCC170 datasets (Student’s *t*-test), respectively. Dataset HCC53 was excluded from the functional enrichment analysis because it had few DEGs. The overexpressed genes in the high-risk group of three datasets were all significantly enriched in cancer-related pathways, such as cell cycle and DNA replication pathways. By contrast, the underexpressed genes in the high-risk group of three datasets were all significantly enriched in metabolism-related ways, including fatty acid degradation and carbon metabolism (FDR < 5%, Figure 3A). These findings indicated that the high-risk patients with poor prognosis were considerably associated with faster proliferation ability and suppressed metabolism, which was in line with a previous report [35].

Désert et al. reported that a non-proliferative PP subtype in HCC has the lowest potential to recur [28]. Compared with the high-risk group of HCC197, the proportion of samples classified as the PP subtype in the low-risk group was predominantly higher (44/64 versus 54/133, Fisher’s exact test, *p* < 0.001, Figure 3B). Consistent results were observed in the other three datasets (Appendix A). In addition, the proliferation ability summarized from the expression levels of 43 proliferation-relevant genes in the high-risk group was significantly higher than that in the low-risk group (Wilcoxon rank-sum test, *p* < 0.05, Figure 3C).

The metabolic characteristics between distinct prognostic groups in the HCC197 dataset were further explored on the basis of the activity of 108 metabolic pathways. The results showed that 67 metabolic pathways, which were divided into seven modules (amino acid-related metabolism, carbon metabolism, drug metabolism, biosynthesis and metabolism of polysaccharides, lipid metabolism, cofactor and vitamin metabolism, and other metabolism), were significantly different between the high- and low-risk groups (limma, FDR < 5%, Figure 3D). Furthermore, compared with the low-risk group, these metabolic pathways in the high-risk group exhibited significantly lower GSVA scores. Additionally, the clinical and pathologic characteristics, except for stage, were equally distributed between the two prognostic subgroups (Figure 3B).

Collectively, these results indicated that samples in the high-risk group exhibited a higher proliferation ability and lower metabolic activity than those in the low-risk group.

### 3.3. Distinct Epigenomic and Genomic Characteristics between the Two Prognostic Groups

In HCC197, 6 and 132 DM sites corresponding to 6 and 126 genes were significantly hypermethylated and hypomethylated in the high-risk group versus the low-risk group (Wilcoxon rank-sum test, FDR < 1%), respectively. Among the above 132 methylated genes, 58 genes overlapped with 3678 DEGs. Strikingly, 47 hypomethylated genes were consistently differentially overexpressed in the high-risk group, including TNFRSF11A and TP73 (Figure 4A).

The CNV analysis showed that the frequencies of copy number gain at 8q24.13 and 8q21.13 in the high-risk samples were significantly higher than the corresponding frequencies in the low-risk patients (76.56% versus 58.02% and 73.44% versus 48.85%, left of Figure 4B). There were 29 genes located in the two amplified regions, of which 27 genes were significantly overexpressed in the high-risk group (Appendix A). Overexpression of *ATAD2* and *PVT1* in *8q24.13*, *RAD54B*, *LAPTM4B*, and *RRS1* in 8q21.13 were reported to be associated with tumor proliferation and progression of HCC [36,37,38,39,40]. Meanwhile, the frequencies of copy number loss at 13q14.2, 13q22.2, and 17p11.2 were 62.50%, 50%, and 46.88% in the high-risk patients, respectively, which were significantly higher than the corresponding frequencies of 40.46%, 27.48%, and 27.48% in the low-risk patients (right of Figure 4B). Interestingly, 11 genes located in the three deleted regions were significantly underxpressed in the high-risk group (Appendix A), including tumor suppressor gene *RB1* in 13q14.2.

Compared with the low-risk group, a higher somatic mutation frequency was observed in the high-risk group. The top 10 genes with the highest mutation rates in the two risk subgroups are shown in Figure 4C. Notably, the driver mutation genes were not consistent between the two prognostic groups. The top six genes in the high-risk group were *TP53*, *CTNNB1*, *TNN*, *PCLO*, *ALB*, and *MUC16*, which all had varied ranks in the low-risk group. The somatic mutation on *TP53* was more prevalent in the high-risk group, which is concordant with the fact that the *TP53* mutation was positively associated with a poor prognosis of HCC [41]. Additionally, compared with the low-risk group, 75 genes had significantly higher mutation frequencies in the high-risk group (Fisher’s exact test, *p* < 0.05). Among them, nine genes that were mutated in at least five samples from either prognostic group are shown in Figure 4D. Moreover, the high mutation frequency of *RB1* in the high-risk group might also contribute to its significant underexpression.

These results suggested that the epigenomic and genomic alternations exerted joint effects on the transcriptional dysregulations between the two prognostic groups, leading to poor prognoses for early HCC patients.

### 3.4. Distinct Immune Landscape for HCC Prognostic Groups

The immunologic landscape between different prognostic groups showed that higher scores of activated dendritic cells (aDCs), human lymphocyte antigen (HLA), MHC class I, macrophages, Th1 cells, Th2 cells, and Treg cells were observed in the high-risk group, while higher scores of B cells, mast cells, NK cells, and type II IFN response were observed in the low-risk group (Wilcoxon rank-sum test, *p* < 0.05, Figure 5A and Appendix A). The higher scores of macrophages and Treg cells, which created an immunosuppressive microenvironment, were both correlated with a poor prognosis of HCC [42,43]. Furthermore, the higher score of type II IFN response in the low-risk group indicated an activated immune microenvironment with NK cells, which might protect against tumor development [44].

Then, we investigated the relationship between immune-checkpoint genes and the two prognostic groups. Compared with the low-risk group, more immune-checkpoint genes were significantly overexpressed in the high-risk group (Wilcoxon rank-sum test, *p* < 0.05), including *TIM3*, *B7-H3*, *TIGIT*, *TNFSF15*, and *CD44*, which further confirmed an immunosuppressive microenvironment in the high-risk group (Figure 5B and Appendix A).

### 3.5. Distinct Therapeutic Benefits for HCC Prognostic Groups

We further investigated the potential drugs for two prognostic subgroups in the HCC197 dataset. In the 138-drug screen, patients in the low-risk group were predicted to be more sensitive to 40 kinds of drugs, e.g., gefitinib, whereas patients in the high-risk group were predicted to be more sensitive to the other 42 kinds of drugs, e.g., gemcitabine (Figure 5C). It is well known that gefitinib is effective against certain types of cancer. Furthermore, we observed that *EGFR*, the target gene of gefitinib, was significantly overexpressed in the low-risk group (edgeR, log2|Fold Change (FC)| > 1, FDR < 10%). These results were consistent with those of a previous study, which indicated that high expression of *EGFR* was associated with a good response to gefitinib in non-small cell lung cancer [45]. Our results might provide new strategies for therapy in HCC.

Furthermore, a lower TIDE score was observed in patients in the low-risk group (Figure 5D), indicating that they might have a better response to immunotherapy than patients in the high-risk group.

### 3.6. Validation of the 10-GPS in the Institutional Transcriptomic Data and Public Proteomic Data

The prognostic performance of the 10-GPS was further evaluated in the 47 stage I–II samples generated in our previous study [24]. A total of 29 patients were classified into the low-risk group by the 10-GPS, which was associated with moderately good RFS and significantly long OS (Figure 6A). Moreover, the 10-GPS was able to predict the RFS and OS in the 72 samples (Appendix A). These results further demonstrate that the 10-GPS can predict the prognosis of early HCC after surgery.

The proteomic data of 101 stage I–II HCC patients in the CPTAC data portal helped us to validate the 10-GPS at the proteomic level. In the proteomic data, 11 proteins encoded by the signature genes in the 10-GPS were measured and constructed into six protein pairs. Similarly, according to the half-voting rule, a patient was classified into the high-risk group when at least three protein pairs voted for high risk; otherwise, they were classified into the low-risk group (Figure 6B). The six protein pairs divided 40 patients into the high-risk group and 61 patients into the low-risk group, and the two groups had significantly different DFS (HR = 2.56, 95% CI: 1.1–5.97, *p* = 0.024, C-index = 0.623) and OS (HR = 2.30, 95% CI: 1.12–4.74, *p* = 0.02, C-index = 0.604) (Figure 6C). The results suggest that the 10-GPS can perform well in the proteomic data, which might facilitate its routine implementation in clinical settings.

## 4. Discussion

In this study, an individual prognostic signature consisting of 10 gene pairs with 16 metabolism-related genes, named the 10-GPS, was developed to predict clinical outcomes of early HCC. The signature was validated in multiple independent transcriptomic and proteomic datasets. Patients in the high-risk group had significantly shorter survival than patients in the low-risk group. Compared with the low-risk group, the high-risk group was characterized by lower metabolic activity, higher proliferation abilities, lower immune cells infiltration, and less immunotherapy benefit. The multi-omics analysis showed that the epigenetic and genomic alternations could corporately contribute to the transcriptional differences between the two prognostic groups. The 10-GPS was further prospectively validated in 47 institutional transcriptomic samples and 101 public proteomic samples.

At present, the treatments for early HCC and advanced HCC are considerably different. Surgical resection is the first choice of treatment for early (TNM stage I–II) HCC patients whose liver function is well-preserved but not for advanced HCC with high malignancy. Nevertheless, approximately 60–70% of patients suffer from relapse within 5 years, which is the main factor for poor prognosis. Previously, Nault et al. identified a five-gene score from I–IV stage HCC samples to predict the prognosis of HCC patients after resection [10], which might result in irrelevant features in the prognosis of early HCC. In this study, we aimed to develop a risk stratification model to predict the clinical outcome of early HCC patients after surgery, and only TNM stage I–II samples with resection were collected and analyzed. Univariate and multivariate Cox regression analysis confirmed that the 10-GPS remained an independent predictor after adjusting for clinical variables, e.g., stage and histologic grade. In addition, treatment history before hepatic resection is a potential factor for the prognosis. Although 197 stage I–II samples without any treatment before surgery from TCGA were selected as the discovery dataset, the treatment histories for samples in HCC170 and HCC53 were not unknown. The influence of treatment history on the risk stratification model for early HCC prognosis needs to be further evaluated.

We also noticed that the robustness of the 10-GPS was reduced when most genes in the signature were not expressed or detected. Our previous study and the present study revealed that the gene pair signature can be robust when more than 60% of gene pairs are detected [16]. This study also showed that 60% of protein pairs that consisted of proteins encoded by the signature genes achieved good performance at the proteomic level. However, the impact of the gene pair signature needs further assessment, and the generalization of the signature needs to be further improved and optimized in clinical application.

ICI therapy is a promising approach to improve survival in several cancers, and some immune-checkpoint proteins, e.g., *CTLA-4*, *PD-1*, and *PD-L1*, are viewed as potential biomarkers of ICI response. Patients with high expression of immune checkpoints are more likely to respond to immunotherapy [46]. Nevertheless, conflicting results were obtained in subsequent studies [47]. Zhang et al. showed that not all patients with high expression of *PD-L1* responded to anti-PD-L1 therapy [48]. A recent study also reported that there was no relationship between *PD-L1* expression and treatment outcome in HCC [49]. The mechanism of immune checkpoints as biomarkers for immunotherapy is not fully understood. As a comprehensive index, TIDE mimics the dysfunction and exclusion of immune cells, which might serve as a more reliable alternative biomarker for predicting the ICI response [33]. In this study, the TIDE score did not indicate that patients in the high-risk group with more immune checkpoints with high expression had a better response to immunotherapy. The complex immune microenvironment and the interaction with tumor cells might explain the conflicting results. The differences in tumor immune infiltration and the immunotherapy benefit between the two groups need further investigation in bulk and single-cell sequencing data. The findings regarding the immune landscape in Figure 4 also need to be confirmed by experiments, e.g., by IHC.

We noticed that four genes, *EZH2*, *SETD1B*, *PGS1*, and *G6PD*, appeared twice in the 10-GPS, and they might play important roles in determining the clinical outcome. Specifically, *G6PD* is the rate-limiting enzyme of the pentose phosphate pathways, which is elevated in many cancers to promote tumor growth [29]. Our results showed that *G6PD* was consistent significantly overexpressed in the high-risk group in all four transcriptomic datasets (Appendix A). The protein level of G6PD was also higher in the high-risk group than in the low-risk group. The Kaplan–Meier curves showed that high expression of *G6PD* was significantly associated with short PFI, RFS, and OS times in all four transcriptomic datasets (Appendix A). Similar results regarding G6PD were observed in the proteomic data of HCC101 (Appendix A). These results are in concordance with those of a previous study [50] and suggest that *G6PD* is a potential prognostic biomarker for HCC.

## 5. Conclusions

In summary, this study proposes an individual prognostic signature consisting of 16 metabolism-related genes that can accurately evaluate the outcome of early HCC and characterize inter-tumor heterogeneity in HCC.

## Figures and Tables

**Figure 1 cancers-14-03957-f001:**
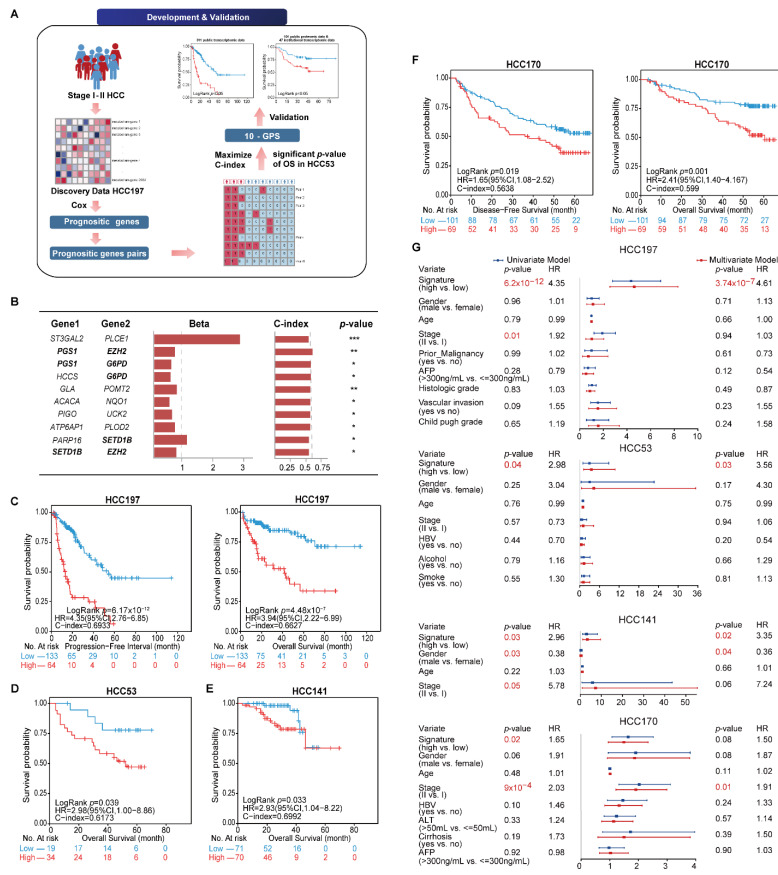
Identification and validation of 10-GPS in transcriptomic datasets. (**A**) The general flowchart of this study. (**B**) The 10-GPS for risk stratification. Genes shown in bold appeared twice in the signature. The univariate Cox regression model calculated the Beta and *p*-value. Beta was the risk coefficient of the REO of gene pair in 10-GPS, where Beta > 0 indicates that *E*_1_ < *E*_2_ is a risk factor and vice versa. *, **, and *** indicate *p* < 0.05, *p* < 0.01, and *p* < 0.001, respectively. (**C**,**D**) The 10-GPS predicted the PFI and OS of two prognostic groups in datasets HCC197 and HCC53. According to the half-voting rule, a patient was classified into the high-risk group (red line) if at least five gene pairs voted for high risk; otherwise, the patient was classified into the low-risk group (blue line). (**E**,**F**) The DFS and OS of two prognostic groups in HCC141 and HCC170 were independent validation datasets. (**G**) Univariate and multivariate Cox regression analyses for the 10-GPS in the discovery and validation datasets.

**Figure 2 cancers-14-03957-f002:**
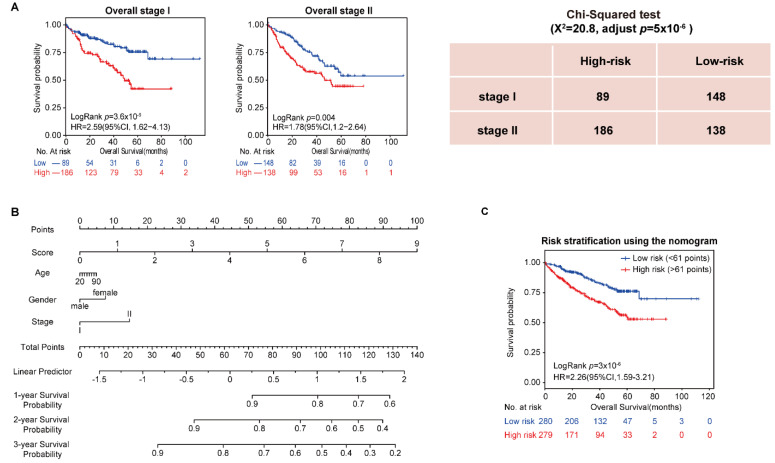
Analysis of 10-GPS in overall stage I–II samples. (**A**) OS of two prognostic groups in overall stage I and overall stage II samples. (**B**) Development of a composite nomogram to predict 1-year, 2-year, and 3-year survival probability. The nomogram was constructed on the basis of 10-GPS, age, gender, and stage in overall HCC samples. (**C**) The survival difference between patients divided by the median of the composite score.

**Figure 3 cancers-14-03957-f003:**
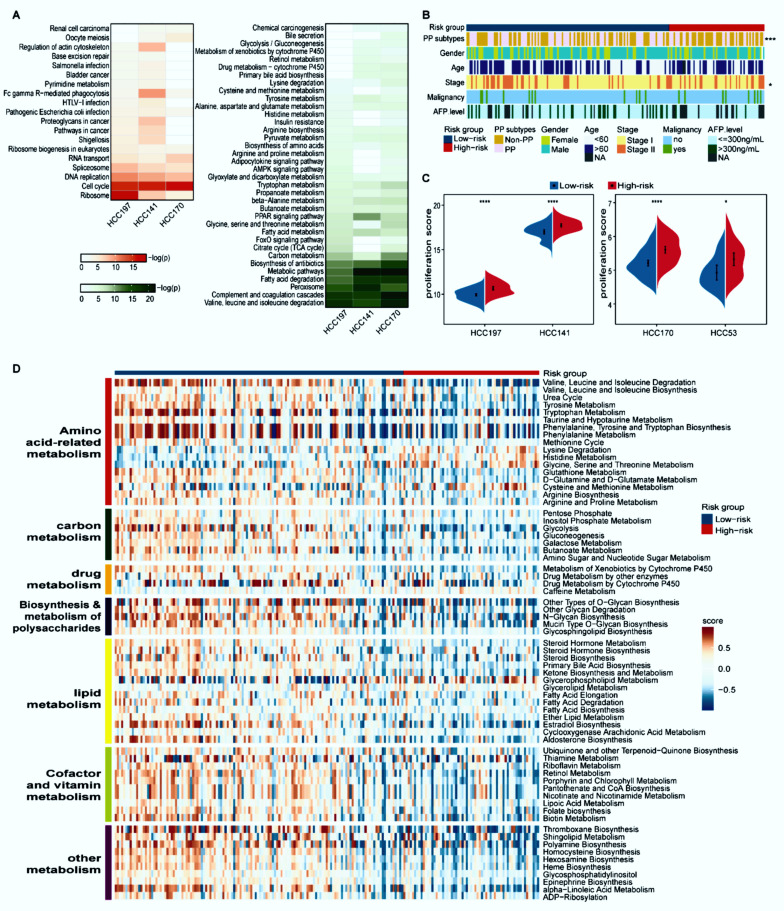
Distinct proliferation and metabolism characteristics of the two prognostic groups. (**A**) Functional enrichment analysis of DEGs between the two prognostic groups. Red and green respectively represent pathways enriched by the differentially overexpressed and underexpressed genes in the high-risk group compared with the low-risk group. The *p*-values were adjusted by BH (FDR < 5%). (**B**) The distribution of PP and non-PP subtypes, age, gender, stage, AFP level, and malignancy along with two prognostic groups. Patients were classified into the PP and non-PP subtypes according to the median value of the principal component analysis (PCA) of eight representative proliferation genes; * *p* < 0.05, *** *p* < 0.001. (**C**) The split violin plots of the average expression levels of 43 proliferation-associated genes. * *p* < 0.05, **** *p* < 0.0001. (**D**) Heatmap of 67 differential metabolism pathways. Blue and red bars represent low- and high-risk samples, respectively. The activity scores were quantified by GSVA using DEGs in each pathway and compared by limma with *p* < 0.05.

**Figure 4 cancers-14-03957-f004:**
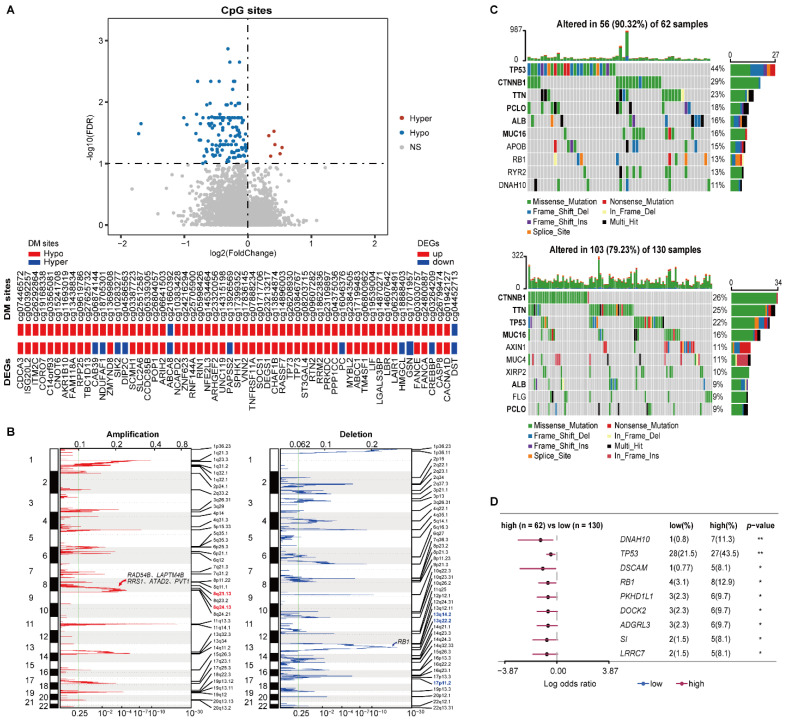
Multi-omics characteristics of the two prognostic groups. (**A**) The volcano plot of the CpG sites (the top pane) and the concordance of DM sites with DEGs (the bottom panel). Wilcoxon rank-sum test with FDR < 1% was used to identify the DM sites between two prognostic groups. (**B**) The CNV alterations of samples. The green line represents the cut-off point that determined the significance, *q*-value < 0.25. (**C**) The top 10 genes with the highest mutation rates in the high-risk (top) and low-risk (bottom) groups. (**D**) Genes with significant mutations in at least five samples from either prognostic group. Fisher’s exact test was used to calculate *p*-values. * *p* < 0.05, ** *p* < 0.01.

**Figure 5 cancers-14-03957-f005:**
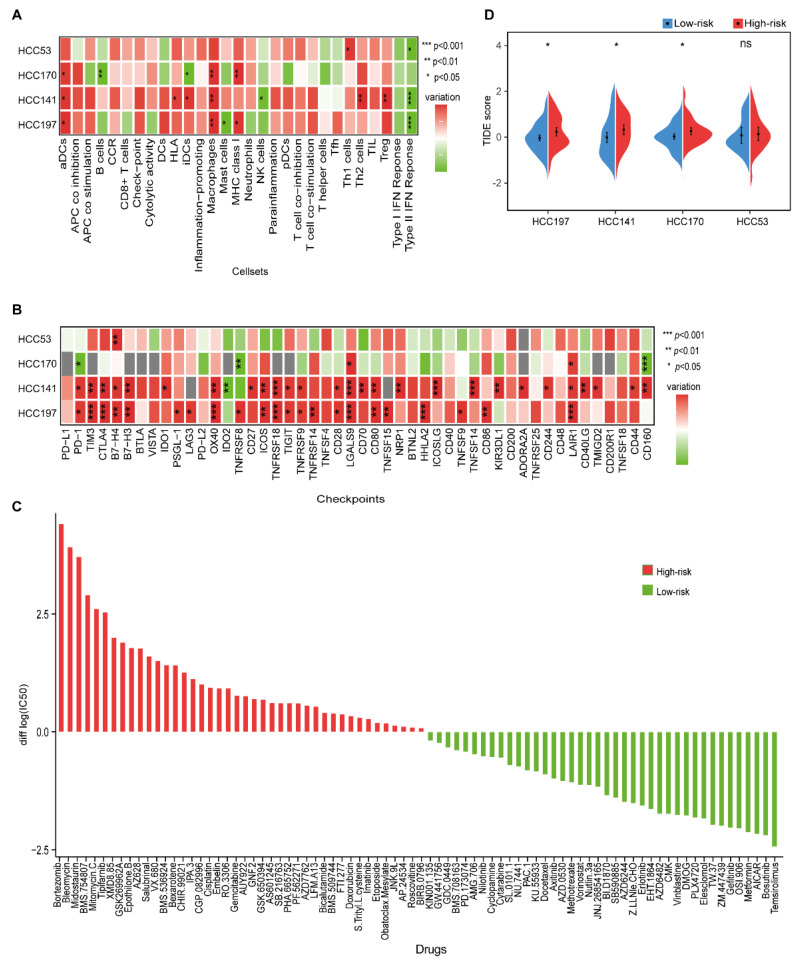
The immune characteristics and therapeutic benefits of two prognostic groups. (**A**) Heatmap of the levels of immune cell infiltrates in four datasets. The differences in ssGSEA scores between the high- and low-risk groups were calculated using the Wilcoxon rank-sum test. (**B**) The expression levels of 48 immune-checkpoint genes between the high- and low-risk groups in four datasets. Red and green represent a higher expression in the high- and low-risk groups, respectively. *, **, and *** indicate *p* < 0.05, *p* < 0.01, and *p* < 0.001, respectively. (**C**) Estimated drug sensitivity for patients in the high- and low-risk groups. The horizontal axis represents drugs, and the vertical axis represents the difference in the median of log_2_|IC50| values between the high- and low-risk groups. Red and green represent the drugs that might be more sensitive in the high- and low-risk groups, respectively. (**D**) Split violin plot of the TIDE scores for patients in the high- and low-risk groups in the four datasets.

**Figure 6 cancers-14-03957-f006:**
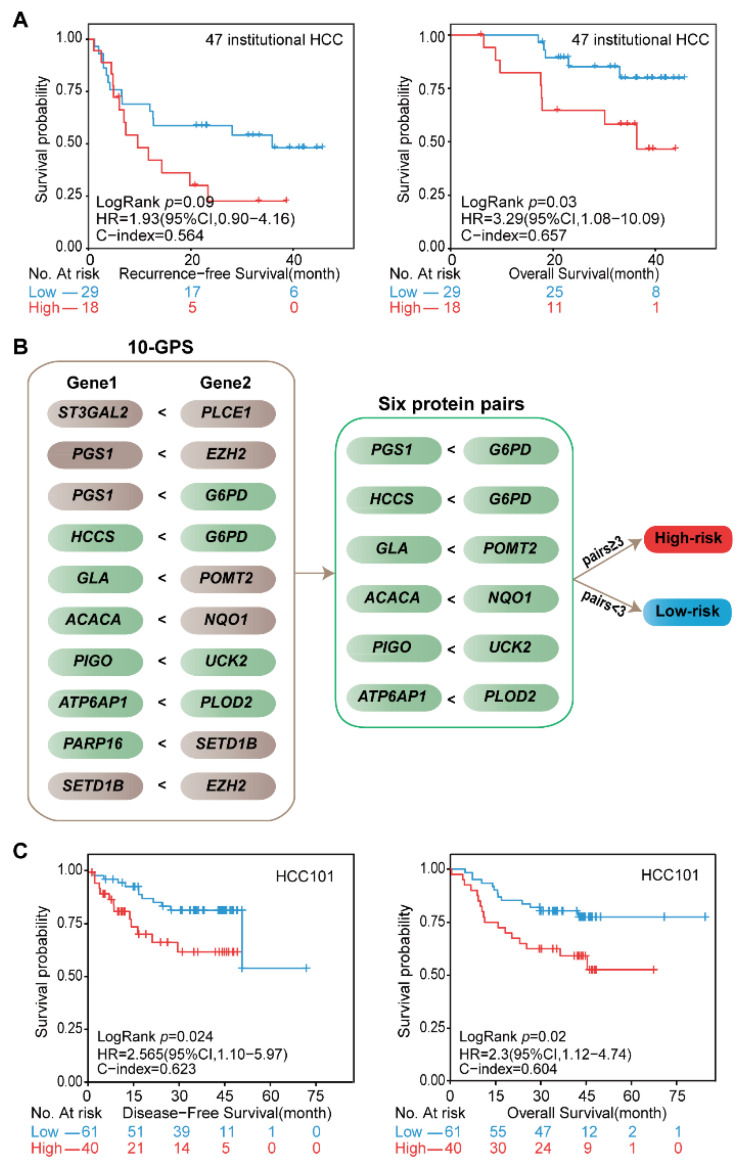
Validation of the signature at institutional transcriptomic data and public proteomic data. (**A**) Kaplan–Meier curves of the RFS and OS for the 47 institutional transcriptomic HCCd. (**B**) Six protein pairs in the proteomic data. Green indicates that the protein was detected in the proteomic data, and gray indicates that the gene was not detected in the proteomic data. (**C**) Kaplan–Meier curves of the DFS and OS for the proteomic data of HCC101. According to the half-voting rule, a patient was classified into the high-risk group (red line) when at least three protein pairs voted for high risk; otherwise, they were classified into the low-risk group (blue line).

**Table 1 cancers-14-03957-t001:** Description of the data used in this study.

	Discovery		Validation		Institutional Validation	
	HCC197	HCC53	HCC170	HCC141	HCC101	HCC47
Accession	TCGA	GSE116174	GSE14520	LIRI-JP	CPTAC	HRA000464
Platform	Illumina Hiseq	GPL13158	GPL3921	Illumina Hiseq	Whole Proteome	Illumina Hiseq
Survival	PFI and OS	OS	DFS and OS	OS	DFS and OS	RFS and OS
Country	Mix	China	USA	Japan	China	China
Sample size	197	53	170	141	101	47
Age						
≥60	104	19	37	114	40	22
<60	93	34	133	27	61	25
Gender						
Male	142	47	143	96	77	37
Female	55	6	27	45	24	10
TNM stage						
I	138	8	93	36	87	24
II	59	45	77	105	14	23
AFP						
>300 ng/mL	34	-	66	-	64	9
≤300 ng/mL	132	-	101	-	37	38
Cirrhosis						
Yes	-	-	153	-	71	29
No	-	-	17	-	30	17
NA						1
Viral infection						
HBV	78	38	165	-	101	21
HCV	28	0	-	-	0	
HBV/HCV	5	0	-	-	0	
NA	86	15	5	-	0	26
Histologic grade						
G1/G2	118	-	-	-	-	-
G3/G4	78	-	-	-	-	-
NA	1	-	-	-	-	-
Vascular invasion						
Yes	51	-	-	-	-	-
No	134	-	-	-	-	-
NA	12	-	-	-	-	-
Child-Pugh						
A	146	-	-	-	-	-
B/C	10	-	-	-	-	-
NA	41	-	-	-	-	-

**Table 2 cancers-14-03957-t002:** The comparison of demographics and clinical features among datasets.

Variable		HCC197 (n = 197)	HCC53 (n = 53)	HCC170 (n = 170)	HCC141 (n = 141)	*p*-Value
Age	≥60	104	19	37	114	*p < 2.2* × 10^−^^16^
Gender	Male	142	47	143	96	*p <* 5.8 × 10^−^^4^
TNM stage	I	138	8	93	36	*p < 2.2* × 10^−^^16^
	II	59	45	77	105	
AFP	>300 ng/mL	34	-	66	-	*p < 1.9* × 10^−^^4^
Viral infection	HBV	78	38	165	-	*p <* 2.3 × 10^−^^15^

Note. Chi-square test was used for all variables but viral infection. Fisher’s exact test was used for viral infection variable.

## Data Availability

Publicly available datasets were used in this study. These can be found in TCGA, ICGC, CPTAC repository, and GEO at GSE14520 and GSE116174. The institutional transcriptomic data of the 72 samples are available in the GSA repository with accession number HRA000464. Access to HRA000464 is restricted because genetic data is personally identifiable. The data and clinical information will be shared upon reasonable request to the corresponding author.

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
