# Peer review of "Metabolism-Related Gene Pairs to Predict the Clinical Outcome and Molecular Characteristics of Early Hepatocellular Carcinoma"

_cancers, 2022, doi:10.3390/cancers14163957_

Round 1
Reviewer 1 Report
Report
The manuscript entitled “Metabolism-related gene pairs to predict the clinical outcome and molecular characteristics of early hepatocellular carcinoma” submitted by Junling Wu et al highlights the prognostic significance of nomogram constructed using 10-gene pairs (GPS), related to metabolism. The metabolism-related gene signatures developed in this study can potentially impact clinical management of HCC patients by improving the prognosis, stratification of patients individually and by predicting the therapeutic benefits. After stratification of different prognostic groups, the immunologic landscape between different prognostic groups showed that higher scores of activated dendritic cells human lymphocyte antigen (HLA), MHC class I, macrophages, Th1 cells, Th2 cells, and Treg cells were associated with the high-risk group, while higher scores of B cells, mast cells, NK cells and type II IFN response were observed in the low-risk group. This study proposed individualized patient predictive and prognostic signatures that can potentially impact evaluation of clinical outcome of HCC patients.

Author Response
We thank the reviewer for all comments.
Reviewer 2 Report
Thank you for the opportunity to review the article “Metabolism-related gene pairs to predict the clinical outcome and molecular characteristics of early hepatocellular carcinoma” by Wu et al. Prognostic strategies for recurrence-free survival after HCC resection or liver transplantation are largely based upon pre-surgical biomarker levels and explant staging. Although these approaches can help guide surveillance strategies, our understanding of the molecular mechanisms and characteristics of aggressive early-stage HCC with increased recurrence risk is limited. In this article, the authors investigate metabolic gene signatures and their ability to stratify a variety of survival outcomes in early-stage HCC.
Overall
My overall concern relates to limitations in the dataset. Although these limitations are likely unavoidable, they should be discussed and developed in the manuscript as potential limitations of the study. Please see the detailed comments below. In summary, the clinical reality of the approach is still tethered to post-resection specimen analysis. Although genetic and molecular approaches will hopefully revolutionize patient outcomes through refining prognosis at various stages from diagnosis to surgical intervention, we need them to outperform current clinical indicators and control for these factors. Ideally, these indicators including AFP, iterations of tumor burden (Milan Criteria, small / intermediate / large HCC), treatment history / treatment effect, pathological grading, pathological staging, lymphovascular invasion, satellitosis, and cirrhosis staging are present and controlled across all cohorts for a harmonized data analysis. The author’s approach is thorough and commendable, but a more thorough description of the study limitations and how they may influence the study outcomes and interpretation is required.
General Comments
- The manuscript employs multiple time-to-event designations that all appear to be recurrence-free survival (progression-free interval). This and overall survival are the two main time-to-event metrics in the manuscript. The authors should choose a single designation for overlapping metrics and employ it consistently throughout the manuscript.
- Given the manuscript focus on recurrence, it would be important to know the fraction of T2 patients with vascular invasion and how grouping out this fraction affected overall performance of the gene panel. Vascular invasion is an extremely significant prognostic factor for recurrence risk that falls under T2 staging and is a major distinguishing factor to T1a, T1b, and T2 multifocal without vascular invasion. For instance, using the BCLC staging system, surgical intervention would be contraindicated in HCC with known or suspected vascular invasion. The authors partially address this issue in Figure 2A, by controlling for TMN stage, although without thorough accounting of other known risk factors.
- The role of underlying cirrhosis is challenging to interpret in the dataset. Cirrhosis status is unknown in a little more half the overall dataset and the dataset is heavily weighted toward HBV-HCC, which can have a higher frequency of HCC incidence absent cirrhosis and with well-preserved liver function. However, where cirrhosis data is available, the frequency of cirrhosis is extremely high (80%). If the study was specifically focused on recurrence-free survival, the OS risk associated with an unknown fraction of Child-Pugh B, or greater, patients in the dataset may be properly controlled. However, the manuscript uses recurrence-free survival and OS in tandem. The influence of Child-Pugh B cirrhosis on HCC resection outcomes may be substantial and warrants discussion.
Specific Comments
Methods
- The cohort is consistently referred to as Stage I – II. The methods should specify this is according to the TMN staging system.
- A status of treatment naïve prior to hepatic resection was acknowledged for the institutional cohort (HCC47). From the text, I presume the treatment history in the other cohorts is unknown. Treatment history would be a potentially significant factor in a molecular analysis of post-surgical specimens and warrants discussion as a limitation of the dataset.
- What is the rationale for thresholding AFP at 300 ng/mL? Current data would argue that AFP in the 50 – 300 ng/mL range is a significant risk factor for post-surgical recurrence. Please provide a justification for the AFP threshold.
- Section 2.3. This section mentions relapse-free survival data for the HCC47 cohort in Table 1. This data is not included in Table 1. Additionally, it is unclear what the authors define as the randomization for relapse-free survival? I believe recurrence-free survival, or progression-free interval as used in other instances in the text, would be a more accurate description of the time-to-event data measurement.
- Section 2.5. In identifying the metabolism-related gene pairs of interest, the authors selected genes using univariate Cox proportional hazards for pairs associated with both tumor recurrence and death. As stated, “tumor recurrence and death” could be interpreted as a derived, time-independent dichotomous outcome. This section of the methods should be revised for clarity. Later in the results section, the authors state that progression-free interval (PFI) was used to identify the gene pairs of interest and in a separate cohort the genes were cross-validated for overall survival. This description of the methods / approach provides more clarity than the description in the methods section. The methods / approach should be stated consistently and harmonized in both sections of the article.
Figures and Tables
- Table 2 is very busy. From the text, it appears the purpose of the table is to establish that the cohorts were highly diverse. This could be more effectively and concisely demonstrated using the Chi-square test among the 4 groups? The intergroup Fisher’s exact test would then not be necessary, as there is no further development of the intergroup differences between specific cohorts in the manuscript. If the authors feel the Fisher’s results are required, it is recommended that the reported P values be condensed to ease interpretation. All P values < 0.001 could be reported using as such “p < 0.001”.
- Figure 1. While the signature holds in the multivariate for each cohort, the effect is most exaggerated in the discovery cohort. This is a critical issue because the frequency of vascular invasion for each cohort is not reported in Table 1. I am concerned that if each individual cohort had a robust control over the well-established risk factors for post-surgical HCC recurrence (lower AFP threshold, vascular invasion, pathological staging, Milan Criteria, Child-Pugh, treatment history and treatment response), the effect of the gene signature would be diminished in the multivariate model.
Results
- Section 3.2. An imbalance in proliferative capacity with metabolic derangement is a hallmark of poorly differentiated carcinomas. The finding that these pathways are consistent with the high-risk gene panel group are interesting and could be controlled with explant tumor grade. My concern is that if tumor grading were controlled, the gene panel may fall out of multivariate analysis through the argument that the tumor grade is a product of the underlying cancer genetics and the resulting transcriptomic changes captured by the gene panel. This is an important discussion point as tumor grading is post-surgical standard care at many institutions.
Author Response
We thank the reviewer for all comments. Following is our point-by-point responses to the reviewer’s comments and critiques have been accommodated fully in various parts of the revised version (shown in red color).
